

# Pollination efficacy of stingless bees, *Tetragonula pagdeni* Schwarz (*Apidae: Meliponini*), on greenhouse tomatoes (*Solanum lycopersicum* Linnaeus)

Kanyanat Wongsa[1], Orawan Duangphakdee[2] and
Atsalek Rattanawannee[1]

[1] Department of Entomology, Faculty of Agriculture, Kasetsart University, Chatuchak, Bangkok, Thailand

[2] Native Honeybee and Pollinator Research Center, Ratchaburi Campus, King Mongkut's University of Technology Thonburi, Thung Khru, Bangkok, Thailand

Corresponding author
Atsalek Rattanawannee,
fagralr@ku.ac.th

## ABSTRACT

The stingless bee *Tetragonula pagdeni* is distributed over a vast Southeast Asian territory. This species is commonly used as a commercial insect pollinator. Pollination efficacy of *T. pagdeni* was investigated with tomato (*Solanum lycopersicum* L.) cultivated in greenhouse environments. In the first experiment, the number of fruit sets, number of seeds, fresh weight, and fruit height were quantified in the greenhouse with stingless bees, without stingless bees, and with pollination by mechanical vibration by hand. In the second experiment, the treatments were conducted with tomatoes of indeterminate growth in the greenhouse with and without stingless bees to prevent variation among the different tomato plants. The obtained results showed that a greenhouse with stingless bees presented 85 ± 4.24 fruits per 100 flowers, more than a greenhouse with mechanical vibration (79.5 ± 2.12 fruits per 100 flowers) or a greenhouse without stingless bees (15 ± 0.00 fruits per 100 flowers). In addition, fruit produced in a greenhouse with stingless bees showed greater fruit weight and number of seeds than fruit produced in a greenhouse without stingless bees or pollinated by mechanical vibration. According to the obtained results, we suggest that *T. pagdeni* could be beneficial as an insect pollinator of greenhouse tomatoes in tropical regions, where the use of honeybees and bumblebees would be more difficult.

## INTRODUCTION

Tomato (*Solanum lycopersicum* L.) flower is self-compatible, but releasing the pollen grains through the apical flower requires vibration of the poricidal anthers (*Moura-Moraes et al., 2021*). In an open field, wind action and insect visitation can produce the vibration needed to release pollen grains (*Gaglianone et al., 2018*). However, the use of pollinating agents or actions is required under greenhouse conditions to produce sufficient fruit sets and fruit quality (*Morandin, Laverty & Kevan, 2001*; *Palma et al., 2008*). Mechanical

vibration of tomato flowers is one option for pollination in a greenhouse, but this procedure is expensive and labor intensive and can destroy flowers (*Ilbi & Boztok, 1994*). Another option for pollination of tomatoes cultivated in a greenhouse is the use of bees. There have been attempts to compare the effectiveness of traditional vibration (mechanical vibration) with pollination by honey bees (*Apis mellifera* L.) or bumble bees (*Bombus terrestris*) in greenhouse tomatoes. The results show that bumble bees were effective on tomatoes in greenhouses, whereas honey bees were not (*Sabara & Winston, 2003*). Bumble bees enhance pollination efficacy by increasing pollen grain release using vibration of buzzing behavior on the poricidal anther of tomato flowers (*Free, 1993*; *Moura-Moraes et al., 2021*). Additionally, it is possible that tomato flowers does not attract honey bee visitors because they do not provide nectar (*dos Santos et al., 2009*). The pollination effectiveness of the stingless bee (*Melipona quadrifasciata*) and honey bee (*Apis mellifera*) were tested and compared in tomato greenhouses. Stingless bees were significantly more effective than honey bees in pollinating greenhouse tomatoes. In addition, under greenhouse conditions, the quality of tomato fruits can be increased by the use of stingless bee species with different flower-visiting behaviors (*dos Santos et al., 2009*; *Moura-Moraes et al., 2021*). In temperate regions, many species of bumble bees have been successfully used for greenhouse tomato pollination, but the usefulness of bumble bees is restricted because they are not perennial. Therefore, new colonies are needed each year, which raises the cost. Additionally, in tropical areas, the temperature inside a greenhouse usually exceeds that in the outside environment. This microclimate may thus induce heat stress in bumble bees. Among the numerous bee species native to the tropics, stingless bees have gained attention as potential pollinators of both open-field and greenhouse cultivars.

Stingless bees are eusocial insects that are widely distributed across subtropical and tropical regions (*Quezada-Euán, 2018*; *Rattanawannee & Duangphakdee, 2019*). These groups of eusocial bees, with approximately 500 valid species (*Quezada-Euán, 2018*), are diverse in external morphology, body size, colony size, and foraging strategies (*da Silva et al., 2017*; *Hrncir & Maia-Silva, 2013*). Stingless bees are the major group of pollinators for many native and cultivated plant species in the tropics (*Heard, 1999*; *Michener, 2007*; *Momose et al., 1998*). The advantages of stingless bees as pollinators are floral constancy, populous and consistent perennial colonies, nonfunctional sting, ease of handling, and marked worker recruitment behavior (*Bartelli, Santos & Nogueira-Ferreira, 2014*; *da Silva et al., 2017*). Recently, the management of various agricultural plants in greenhouses with the introduction of different species of stingless bees has been tested. Two stingless bee species (*Scaptotrigona* aff. *depilis* and *Nannotrigona testaceicornis*) were found to be successful in pollinating strawberry greenhouses in Brazil (*Roselino et al., 2009*).
In Malaysia, the stingless bee *Tetragonula iridipennis* was found to be a successful pollinator in cucumber greenhouses (*Mitta et al., 2017*). *Atmowidi et al. (2022)* investigated the pollination effectiveness of stingless bees of two genera, *Tetragonula* in strawberry and *Heterotrigona* in melon, in a greenhouse. *Tetragonula* increased the number of fruits and quality of strawberries, and *Heterotrigona* did the same for melons. Therefore, stingless bees may be an alternative to honey bees and bumble bees as pollinators of economic crops in tropical areas.

At least 33 species of stingless bees have been reported in Thailand (*Attasopa et al., 2018*; *Engel & Rasmussen, 2017*; *Klakasikorn et al., 2005*; *Michener & Boongird, 2004*; *Rasmussen, 2008*; *Sakagami, Inoue & Salmah, 1985*; *Schwarz, 1939*), and at least six species are commonly managed for commercial meliponiculture (*Rattanawannee & Duangphakdee, 2019*). Of these, *Tetragonula pagdeni* is a small species that is easily transferred to artificial wooden hive boxes. It has been qualified by Thailand's National Bureau of Agricultural Commodity and Food Standards as one of two species (*T. pagdeni* and *T. iridipennis* Smith, 1854) that have been successfully domesticated at the farm scale. Colony populations of this species have several thousand individual female workers headed by a single female queen (*Fatima et al., 2018*). These bees not only are effective pollinators of economically important crops and native plants but also produce high honey yields (*Attasopa et al., 2018*; *Fatima et al., 2018*; *Rattanawannee & Duangphakdee, 2019*). Commercial cultivation in greenhouses, including of tomatoes, is rapidly increasing in tropical areas, so there is great potential for the use of native bee species. In this study, we evaluated *T. pagdeni*'s efficacy in pollination and consequent fruit production of tomatoes grown in a greenhouse.

# MATERIALS AND METHODS

## Study site

This study was performed in the Lam Sonthi district, Lopburi province, Thailand (15°18′ 6″N, 101°21′48″E). The climate is tropical, with an average precipitation of 553 mm and an average temperature ranging from 26–30 °C. The experiments were conducted between August 2020 and March 2021. The colonies of *T. pagdeni* were provided by different commercial beekeepers and kept in the study area for 4 weeks before the start of the experiment. All colonies were kept in standard wooden hive boxes (W × L × H = 20 cm × 30 cm × 15 cm), conventionally used for meliponiculture in Thailand. In addition, all colonies were queen-right and contained similar quantities of honey and pollen storage pots at the beginning of the experiments (*da Silva et al., 2017*). One healthy colony was used for each replicate independently, making a total of four different colonies for both experiments. The tomato plant used was a hybrid (cherry tomato: F1-Hybrid) with a growing period of up to 120 days.

Thirty-day-old seedlings were each placed in a five-liter plastic pot (one plant per pot) that contained 5 kg of substrate. The substrate used in each pot consisted of five shares of forest soil and one of cattle manure supplement, with 20 g of NPK 4-14-8 (*Silva-Neto et al., 2013*). Thirty pots were used in each greenhouse in two replicates to make a total of 60 pots in each greenhouse, and each pot was considered a plot (*Silva-Neto et al., 2013*). Manual irrigation and 20 g of NPK 4-14-8 were delivered with cover fertilization 30 days after transplantation. Additionally, manual evaluation of pests and diseases was performed in this study. Pesticide was not used throughout the experiment.

## First experiment

The greenhouses had flat-arch roofs (size: 3 m × 6 m, lateral height: 3 m, height at the top: 4.5 m). The roof was covered with transparent low-density polyethylene with a sheet

thickness of 0.1 mm. To exchange the heat and humidity inside the greenhouse, the lateral walls were installed with an insect proof net with a stitch dimension of 50 mesh. Each greenhouse was 1.5 m away from the next. Three greenhouses with three treatments were used in this experiment. The treatments were one greenhouse with stingless bees (WSB), one greenhouse without stingless bees (WoSB), and one greenhouse without stingless bees but using hand vibration (WoSBHV: each cordon of plants was vibrated by hand for 5 s between 08:00 and 10:00 am twice a week). The temperature and humidity of each greenhouse were measured during the foraging activity experiment using digital sensors (Aideepen DHT22, sensor size: 22 mm × 28 mm × 5 mm, accuracy: humidity ±2% RH (max ±5% RH); temperature < ±0.5 °C). All three experiments were performed in parallel.

Tomato has a period of flower blooming 45–50 days after seeding. Thirty pots of tomato were grown in each greenhouse. Thirty inflorescences (with $6.00 \pm 1.40$ flowers/inflorescence) were randomly selected for each experiment. All pots were maintained approximately 80–120 days after seeding for fruit production. After fruit harvesting, all plants were executed. The new set of seedlings was generated for the second replication, and in total, 60 plants were used for each experiment.

Stingless bees (*T. pagdeni*) were used in the WSB experiment. Before colony introduction into the greenhouse, the colonies were checked for substantial honey and pollen storage. One colony (≈800–1,200 adult worker bees per colony) (*del Sarto, Peruquetti & Campos, 2005*; *dos Santos et al., 2009*) of *T. pagdeni* in a standard wooden box was introduced into the greenhouse as soon as flowers first bloomed (approximately 45 days after transplantation) and was removed 7 days later. One colony was placed in each greenhouse in this experiment. For the WoSBHV treatment, each cordon of plants was vibrated by hand for 5 s at 08.00 and 10.00 am twice a week for 4 weeks (*Cauich et al., 2004*). For the WoSB treatment, no pollinating activities of tomato plants were conducted throughout the experiment.

To evaluate the pollination efficacy of *T. pagdeni* on tomatoes, 10 flowers of each of 10 tomato plants per treatment (WSB, WoSB, and WoSBHV) were randomly tagged. The number of fruit sets of tagged flowers was determined to evaluate the percentage of fruit sets of each group (*Cauich et al., 2004*; *dos Santos et al., 2009*). The fruit production of each group was individually examined to determine the weight, width and height of the fruit (*Cauich et al., 2004*). Thirty fruits per treatment were randomly chosen and dissected to count the number of seeds. The pollination efficacy of *T. pagdeni* (WSB) was compared to that under the other treatments (WoSB and WoSBHV). Analysis of variance (ANOVA) followed by Tukey's multiple comparison test ($p < 0.05$) was performed to compare the number of fruit sets, number of seeds, fruit weight, fruit diameter, and fruit height. Multiple regression analysis was performed to investigate the relationship between the treatments and the number of seeds, fruit weight, fruit diameter, and fruit height. The statistical analyses were performed using the *R* program (*R Core Team, 2018*).

To examine the foraging activity of *T. pagdeni* in greenhouse conditions, the number of foragers returning to the nest with and without pollen was monitored in each replicate colony. The data were recorded for a 5-min period every hour from 05.30 to 18.00 h for 5 days using an action camera (SJCAM, 4K ultra HD: 1080 pixel: 60 fps). The camera was
positioned 30 cm from the nest entrance, setting aside a nest entrance to avoid disturbing forager activity. The returning foragers with and without a pollen load attached to their corbicula were counted (*Cauich et al., 2004*). The differences between the hours of the day for each type of returning forager were analyzed using the residuals of the chi-square test (*Zar, 1999*). Pearson's correlation coefficient was calculated to quantify the relationship between the foraging activity of the bees and the physical factors (*Cauich et al., 2004*; *Zar, 1999*).

### Second experiment

This experiment was performed in the same greenhouses as the previous experiment. Unlike the first experiment, all treatments were applied to the same tomato plants to prevent the variability of fruit production among the different plants (*Silva-Neto et al., 2013*). The treatments were therefore performed in greenhouses with and without stingless bees. In each greenhouse, 20 inflorescences of tomato were randomly bagged to prevent foragers from visiting the flowers, and another 20 inflorescences were tagged without being bagged. One colony of *T. pagdeni* was introduced into the WSB greenhouse, following the same management procedures performed in the first experiment. The number of flowers in each tagged inflorescence was recorded for comparison with the number of fruit sets, which is the fruiting rate per inflorescence. After the senescence of the tagged flowers, the stingless bee colonies were removed from the WSB greenhouse. Twenty tomato fruits were collected from each greenhouse to compare fruit weight, number of seeds per fruit, fruit height, and fruit diameter, according to the procedure of the first experiment. Analysis of variance (ANOVA) followed by Tukey's multiple comparison test ($p < 0.05$) was carried out to compare the fruiting rate, fruit weight, fruit height and fruit diameter, and the number of seeds from bagged and unbagged flowers in greenhouses with and without stingless bees (*Silva-Neto et al., 2013*). The statistical analyses were performed using the *R* program (*R Core Team, 2018*).

## RESULTS

### Foraging activity of *T. pagdeni* in greenhouse conditions

After we introduced the *T. pagdeni* colony into the greenhouse, we found that the foragers started their activities as early as sunrise (around 6:00 am). The number of incoming and outgoing forager bees was not consistent, peaking around 7 am to 1 pm, then slightly declining, and then stopping after sunset (around 6:00 pm) (Table 1). The highest number of incoming forager bees with and without pollen loads occurred at 10:00 am. (The count of incoming bees with pollen was 19 ± 2.43, and that without pollen was 44 ± 3.49) (Fig. 1). The temperature and humidity inside the greenhouse ranged from 30.5–32.0 °C and 64.13–73.84%, respectively. After the temperature inside the greenhouse became higher than 31 °C (at noon), the number of incoming and outgoing foragers significantly decreased ($p < 0.001$; Fig. 1). Pearson's correlation coefficients of temperature and humidity in the greenhouse with the foraging activity of *T. pagdeni* are shown in Fig. 1. The results show that the number of incoming bees (both with and without pollen load) was negatively correlated with the temperature ($r^2 = -0.191$, $n = 130$; $p < 0.001$) (Fig. 1A)

**Table 1 The mean number (±SD) of incoming bees and outgoing bees at different times between 5:00 am and 7:00 pm (video recorded 5 min/h).**

| Time | 05.00 AM | 06.00 AM | 07.00 AM | 08.00 AM | 09.00 AM | 10.00 AM | 11.00 AM | 12.00 PM | 01.00 PM | 02.00 PM | 03.00 PM | 04.00 PM | 05.00 PM | 06.00 PM | 07.00 PM |
|---|---|---|---|---|---|---|---|---|---|---|---|---|---|---|---|
| Incoming bees | 0 | 5 ± 1.34 | 26 ± 4.76 | 37 ± 6.13 | 54 ± 4.44 | 63 ± 4.71 | 53 ± 17.07 | 25 ± 12.47 | 17 ± 3.94 | 13 ± 2.63 | 7 ± 2.20 | 9 ± 3.37 | 8 ± 1.46 | 5 ± 1.87 | 0 |
| Outgoing bees | 0 | 5 ± 1.96 | 61 ± 5.59 | 71 ± 3.52 | 71 ± 5.03 | 77 ± 4.32 | 62 ± 5.20 | 60 ± 5.08 | 53 ± 6.71 | 29 ± 9.11 | 17 ± 3.58 | 10 ± 3.93 | 8 ± 2.06 | 4 ± 1.65 | 0 |

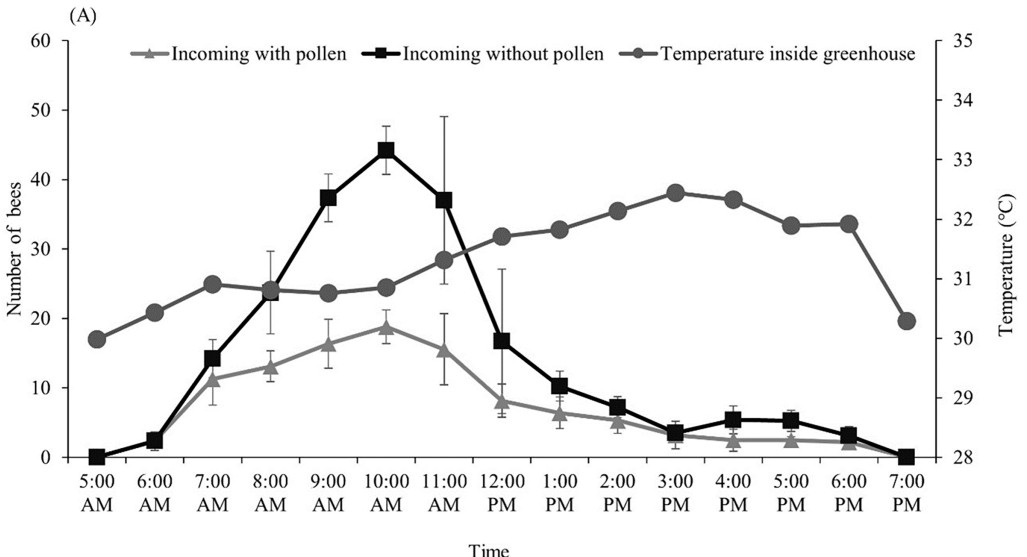

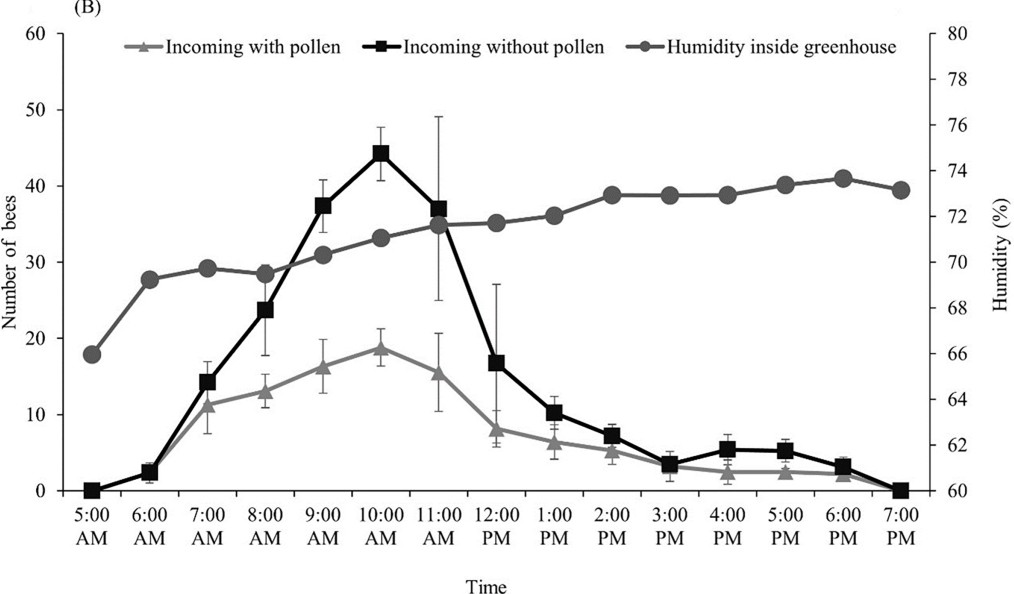

**Figure 1 Average number of *Tetragonula pagdeni* foragers entering the hive with and without pollen in greenhouses.** Plotted against (A) temperature and (B) humidity during 5 min of every hour. Error bars: standard deviation of the mean.

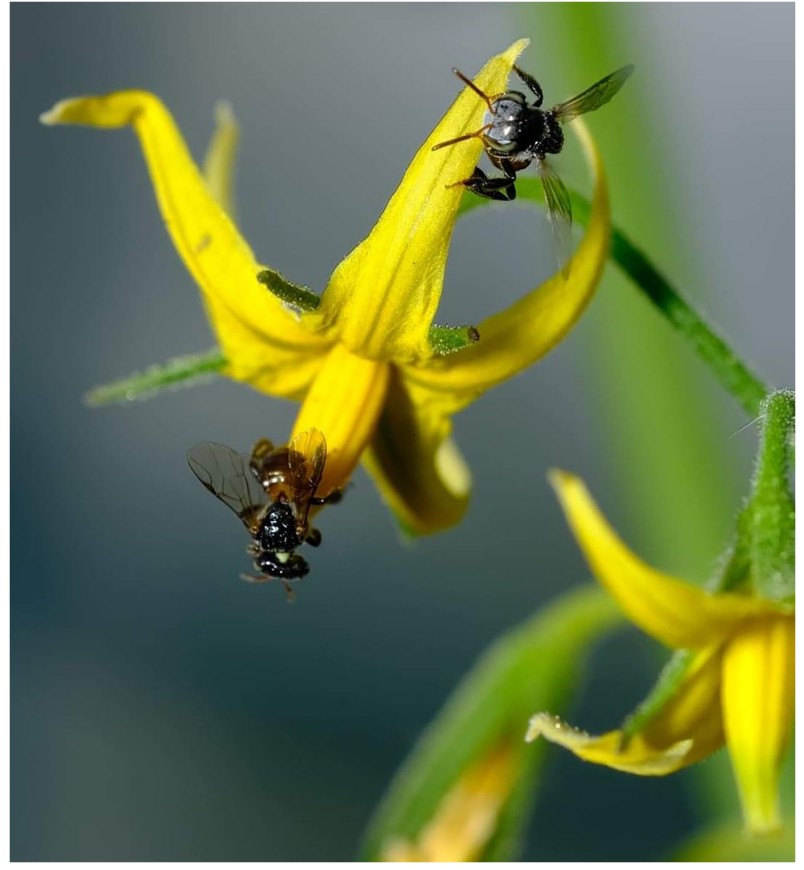

**Figure 2 *Tetragonula pagdeni* workers visiting tomato flowers in a greenhouse.**

and humidity ($r^2 = -0.526$, $n = 130$; $p < 0.001$) (Fig. 1B). Similarly, the number of outgoing bees was negatively correlated with the temperature ($r^2 = -0.208$, $n = 130$; $p < 0.001$) (Fig. 1A) and humidity ($r^2 = -0.517$, $n = 130$; $p < 0.001$) (Fig. 1B).

## First experiment: comparison of the pollination efficacy of stingless bees and hand vibration

We found that *T. pagdeni* foragers visited the tomato flowers in the greenhouses (Fig. 2). The foragers landed on the stamens and crawled over the tube-shaped stamens. Sometimes, the bees extended their tongue toward the base of the anther and calyx of the flower for less than a second and proceeded to stamen observation. The time spent on one flower was relatively short, ranging from 1–3 s. Tomatoes flowers characteristics conceive many of the features that allow buzz-pollinating to extract pollen from their anther (*De Luca & Vallejo-Marín, 2013*). However, no vibrating or buzzing behavior was seen during our experimental observation. Instead, they were walking around on anther cone and some biting into the tip of the cone are notably observed. No data of a different was observed in the percentage of fruit sets between the WSB (85 ± 4.24) and WoSBHV (79.5 ± 2.12) treatments (Fig. 3A). As expected, pollination by *T. pagdeni* and significant mechanical vibration (WSB and WoSBHV) both increased the number of fruit sets compared to the

WoSB treatment (without pollination agents) (Fig. 3A). No data of a different was found in number of fruit set events between WSB and WoSBHV. The tomatoes produced in the WSB treatment showed significantly greater fruit parameter values than those produced in the WoSBHV and WoSB treatments (fruit weight: 4.43 ± 1.03 and 3.20 ± 0.97, $F_{(2,147)} = 86.519$; $p < 0.001$; seed number: 17.75 ± 2.39 and 13.42 ± 2.60, $F_{(2,147)} = 48.489$; $p < 0.001$; fruit height: 27 ± 3.31 and 23.3 ± 3.24, $F_{(2,147)} = 92.374$; $p < 0.001$; and fruit diameter: 16.82 ± 2.43 and 14.28 ± 2.25, $F_{(2,147)} = 13.558$; $p < 0.001$) (Figs. 3B–3E). Additionally, we found that fruit weight was positively correlated with the number of seeds ($r = 0.455$; $p < 0.001$), fruit height ($r = 0.809$; $p < 0.001$), and fruit diameter ($r = 0.654$; $p < 0.001$).

### Second experiment: bagged and unbagged conditions

In greenhouses with stingless bees, the number of fruit sets of the unbagged flowers (90%) was significantly larger than that of the bagged flowers (20%) or flowers in greenhouses without stingless bees (unbagged flowers and bagged flowers were 9% and 2%, respectively) ($F_{(3,76)} = 20.832$; $p < 0.001$) (Table 2). We also found that all fruit parameter values of the unbagged flowers in the greenhouse with stingless bees were higher than those of the bagged flowers and flowers in greenhouses without stingless bees, except for the fruit transverse diameter (Table 2). However, no data of a different was detected in seed number, fruit weight, or fruit diameter between the bagged flowers of the greenhouses with stingless bees and the unbagged flowers of the greenhouse without stingless bees ($p \geq 0.05$).

## DISCUSSION

This study showed that *T. pagdeni* increases the quality and production of tomatoes in greenhouses similarly to those pollinated by hand vibration, but pollination by mechanical vibration is more labor intensive (*Ilbi & Boztok, 1994*). Yet, research has demonstrated that bumble bees produce a higher yield of tomatoes than hand pollination (*Banda & Paxton, 1991*). We therefore suggested that using *T. pagdeni* would be more cost effective. Additionally, the colonies of this stingless bee can potentially acclimate to the conditions inside the greenhouse. We found that the foraging activity of *T. pagdeni* reached its peak between 09:00 and 11:00 each day. The highest number of incoming forager bees with and without pollen loads occurred at 10:00 am, and the activity gradually decreased after 11:00 am. Similar results have been found by *Cauich et al. (2004)*, who investigated the foraging behavior and pollination efficacy of stingless bees (*Nannotrigona perilampoides*). They reported that the number of foraging bees returning to the hive was high between 08:00 am and 11:00 am. *del Sarto, Peruquetti & Campos (2005)* also found that *Melipona quadrifasciata* foraged on tomatoes from 08:00 am through 11:00 am. For the genus *Tetragonula*, the foraging activity of *Tetragonula iridipennis* Smith had a high peak at 10:00 am (8.17 bees/five plant/5 min), and then the activity decreased after 11:00 am (*Painkra & Malllaiah, 2019*). This decline in foraging activity indicates that foraging by stingless bees could be related to anthesis in tomato flowers. A previous study found that the corolla of *S. lycopersicum* opened between 06:00 and 10:00 am (*Teppner, 2005*). For this reason, the foraging activity of stingless bees peaked in the morning and then decreased in

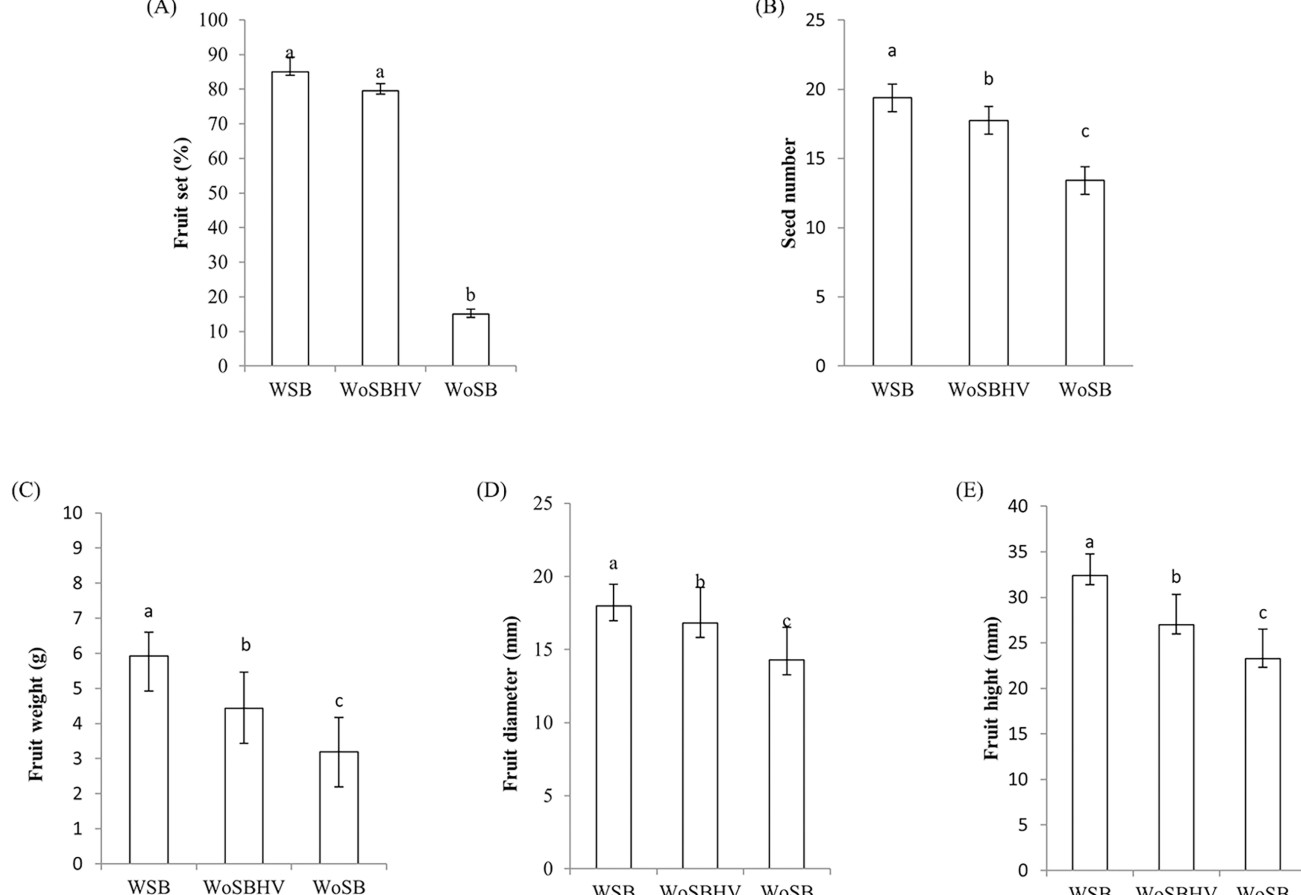

WSB: Greenhouse with stingless bees, WoSBHV: Greenhouse without stingless bees but hand-vibration, WoSB: Greenhouse without stingless bees

**Figure 3 Fruit quality of tomato produced in greenhouse, of the first experiment, with and without stingless bee.** (A) Percentage of fruit set, (B) seed number, (C) fruit weight, (D) fruit diameter, and (E) fruit height. Values are mean ± standard deviation. Different letters above each bar indicate significant differences ($p < 0.05$) by Tukey's pairwise comparison test.  

**Table 2 The percentage of fruit, seed count, fresh weight (g), fruit diameter (mm) and fruit height (mm) of tomato plants (*Solanum lycopersicum*) of the second experiment.**

| Parameters | Greenhouse with stingless bees | | Greenhouse without stingless bees | |
|---|---|---|---|---|
| | Bagged ($n = 20$) | Without bagged ($n = 20$) | Bagged ($n = 20$) | Without bagged ($n = 20$) |
| Number of fruit set (%) | 4 (20)a | 18 (90)b | 2 (10)a | 9 (45)c |
| Number of seeds | 13.45 ± 0.41a | 17.72 ± 0.87b | 5.50 ± 2.83c | 12.48 ± 1.05a |
| Fresh weight (g) | 3.33 ± 1.17a | 5.30 ± 0.06a | 2.68 ± 0.60a | 5.27 ± 2.01a |
| Fruit diameter (mm) | 16.08 ± 2.36a | 17.32 ± 0.34a | 15.63 ± 1.94a | 16.82 ± 0.69a |
| Fruit height (mm) | 22.29 ± 3.48a | 29.21 ± 3.27b | 19.75 ± 0.35a | 24.56 ± 0.94a |

Note:
Means (±SD) followed the different letter in any given row are different from one another ($p < 0.05$) by Tukey's pairwise comparison test. $n$ is the number of inflorescences, and f is the average number of flowers per inflorescence.

the afternoon. However, *Palma et al. (2008)*, using *N. perilampoides* for greenhouse tomato pollination, showed that a decrease in foraging activity was noticeable at 30 °C (*Hikawa & Miyanaga, 2009*; *Peet, Sato & Gardner, 1998*). Interestingly, this gradual decrease corresponded to the decrease in fertile tomato pollen when the average daily temperature was over 30 °C. However, the significant decrease in foraging activity of *T. pagdeni* did not affect the pollination efficacy when compared with hand vibration (Fig. 3).

Considering the quality of fruit produced, *T. pagdeni* produced significantly better fruit than that produced without stingless bees. The weight and number of seeds of the fruits produced in WSB were higher than those in WoSB or WoSBHV. The pollination performance of *T. pagdeni* was confirmed by our second experiment. We found that the tomatoes in greenhouses with stingless bees had a number of fruit sets and fruit quality of unbagged flowers almost twice as high as that found in bagged flowers (Table 2). Similar positive results have been obtained using stingless bees as insect pollinators for some economic crops in a greenhouse or protected environment. For instance, *Cauich et al. (2004)* demonstrated that the use of *N. perilampoides* and hand vibration for pollinating greenhouse tomatoes in tropical climates resulted in significantly higher fruit sets than no pollination. They also reported that the total productivity in kilograms of fruit per square meter was higher in greenhouses with stingless bees than in those with hand vibration and no pollination. Similar results have been found by using *Melipona quadrifasciata* as a pollination agent in greenhouses. *Silva-Neto et al. (2013)* found that tomatoes produced in greenhouses with colonies of *M. quadrifasciata* had 15% more fresh mass and 41% more seeds than those produced in an open environment. The species *M. quadrifasciata* has also been successfully used for greenhouse pollination of other crops, such as eggplant (*Solanum melongena* L.) and sweet pepper (*Capsicum* spp.) (*Cardoso, Salata & Magro, 2015*; *da Silva de Freitas et al., 2015*).

Another important question related to testing the pollination efficacy of *T. pagdeni* in tropical climate greenhouses is the effective transfer of pollen among flowers. This study did not directly determine the removal rate of pollen grains. However, we did find that the number of seeds per tomato fruit produced in greenhouses with *T. pagdeni* was significantly higher than that in greenhouses with hand vibration or greenhouses without stingless bees (Fig. 3). Another indicator of the pollen transfer efficacy of *T. pagdeni* was the visible pollen load attached to the corbiculum of the incoming foragers. These results therefore suggest that *T. pagdeni* shows more efficacy at removing and transferring pollen grains from flowers under greenhouse conditions than what occurs without stingless bees.

Although *T. pagdeni* can increase tomato production in greenhouses, there are some disadvantages of keeping colonies of this species. For example, a colony of stingless bees needs resin as a construction material. This might be more difficult to provide in a greenhouse environment. This problem may be balanced by providing excess cerumen for use by the colony inside the greenhouse (*Slaa et al., 2000*) or by using a healthy colony with encapsulated cerumen before introduction into the greenhouse. A disadvantage of the small size of *T. pagdeni* foragers might be that they have a short foraging range (*Cauich et al., 2004*) and can visit a small number of flowers to obtain pollen (*Slaa et al., 2000*). However, this might be counterbalanced by providing more colonies in each greenhouse.

*del Sarto, Peruquetti & Campos (2005)* determined that a single colony of around 1,200 workers could be enough to pollinate 800–1,500 plants in a confined area such as a greenhouse. A normal queen-right colony of *T. pagdeni* has a population of more than 800 adult workers (AR, unpublished data).

According to our results, we suggest that *T. pagdeni* could be beneficial as an insect pollinator of greenhouse tomatoes in tropical regions, where the use of honey bees and bumblebees would be more difficult. Another advantage of keeping colonies of *T. pagdeni* is the safety of farmers because they do not sting. The species is commonly found in both disturbed and undisturbed habitats. *T. pagdeni* is easily managed using artificial hive boxes and propagation (*Rattanawannee & Duangphakdee, 2019*). Because a colony of *T. pagdeni* is perennial, it is not necessary to replace the hive as would be necessary for bumblebees. More research is needed on the foraging behavioral biology of this stingless bee species and a rigorous investigation of its value as a pollinator compared to currently used bee species.

### Funding

This study was supported by a grant from the Graduate Scholarship as of Fiscal Year 2019 (to support the publication of students' theses in international journals), the Kasetsart University Research and Development Institute (KURDI), the National Research Council of Thailand (NRCT) and Kasetsart University (Grant no. N42A650288), and the Thailand Science Research and Innovation (TSRI) Basic Research Fund: Fiscal year 2021 under project number FRB640008. The funders had no role in study design, data collection and analysis, decision to publish, or preparation of the manuscript.

### Grant Disclosures

The following grant information was disclosed by the authors:
The Graduate School, Kasetsart University.
Kasetsart University Research and Development Institute (KURDI).
National Research Council of Thailand (NRCT) and Kasetsart University: N42A650288.
Thailand Science Research and Innovation (TSRI) Basic Research Fund: FRB640008.

### Competing Interests

The authors declare that they have no competing interests.

### Author Contributions

- Kanyanat Wongsa conceived and designed the experiments, performed the experiments, analyzed the data, prepared figures and/or tables, and approved the final draft.
- Orawan Duangphakdee conceived and designed the experiments, analyzed the data, prepared figures and/or tables, authored or reviewed drafts of the article, and approved the final draft.
- Atsalek Rattanawannee conceived and designed the experiments, performed the experiments, analyzed the data, prepared figures and/or tables, authored or reviewed drafts of the article, and approved the final draft.

## Data Availability

The raw measurements are available in the Supplemental Files.

## Supplemental Information

Supplemental information for this article can be found online at http://dx.doi.org/10.7717/peerj.15367#supplemental-information.

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
