# Peer review of "Pollination efficacy of stingless bees, Tetragonula pagdeni Schwarz (Apidae: Meliponini), on greenhouse tomatoes (Solanum lycopersicum Linnaeus)"

_PeerJ, doi:10.7717/peerj.15367_

## Round 0.1 · original submission · Minor Revisions

Thank you for your submission. I am pleased to accept your article pending minor revisions as suggested by the two expert reviewers. Both reviewers see the valuable contribution of this study to the existing literature base. I think incorporating these changes will make the paper much stronger. I look forward to receiving your resubmission.

·

Basic reporting

Professional English is used throughout. Some of the grammar is a bit awkward, since the authors may not have English as a first language. However, these are very minor points and I feel that their written language is quite acceptable.

Experimental design

The experimental design is adequate and appropriate. I'm glad that the authors included all three treatments of: 1. with stingless bees, 2. without bees, and 3. with mechanical vibration to evaluate tomato fruit set and fruit quality in their greenhouse trials.

Validity of the findings

Only citing the John Free 1993 paper isn't adequate. It is an old paper. Please cite a few recent tomato pollination papers such as those by Peter Kevan and coauthors in Canada. On the other hand, you do cite the papers by dos Santos et al, and Moura-Moraes et al. 2021, Roselyn et al. 2009,
so perhaps this is enough.

Line 54 of manuscript. It is a mistake to say that tomato flowers produce any floral nectar. They do not! Lycoperesicon are pollen only flowers.

Tetragonal pagdeni Schwarz is certainly a pollen robber bee in this circumstance.

I'm surprised that Tetragona pagdeni would readily visit the flowers, but their data show enhanced fruit set when these bee visitors were present. Also, this bee does not buzz pollinate (sonicate) blossoms like other stingless bees (e.g. many Melipona spp. ).

All data appear to have been properly collected and analyzed, with appropriate statistical tests etc.

Conclusions are well stated and derive from the stats etc.

Additional comments

I'm glad that the authors monitored corbiculae pollen loads of returning pollen foragers in the greenhouse trials, documented with a camera. Bees were actively collecting pollen. It is good that the authors described their observational notes on the stingless bees at the flowers Lines 215 to 220). Were they pollen robbing by biting into the flowers? Certainly, they were unlikely to be buzz pollinators of the tomato blossoms. They do state NO buzzing. OK.

Perhaps you might want to cite one or both of these papers:

De Luca, P.A. and Vallejo-Marin, M., 2013. What's the ‘buzz’about? The ecology and evolutionary significance of buzz-pollination. Current opinion in plant biology, 16(4), pp.429-435.

Buchmann, S.L. (1983). Buzz pollination in angiosperms. In: Handbook of Experimental Pollination Biology (eds. Jones, C.E. & Little, R.J). Van Nostrand Reinhold Company, New York, pp. 63-113.

Reviewer 2 ·

Basic reporting

The reader will find it helpful and grasp it well.

Experimental design

The research is clear and intriguing. Also, it's easy for fascinating people to follow the design.

Validity of the findings

Although the research is pretty accurate, it may need to continue and further explore the aspects in order to provide more precise results in the future.

Additional comments

The authors should make any necessary changes based on other suggestions in the content.

Annotated reviews are not available for download in order to protect the identity of reviewers who chose to remain anonymous.

---

## Round 0.2 · accepted · Accept

Thank you to the authors for addressing all the reviewer comments. I am pleased to accept and recommend your paper for publication.